

# The jaw is a second-class lever in *Pedetes capensis* (Rodentia: Pedetidae)

Philip G. Cox

Department of Archaeology, University of York, York, UK
Hull York Medical School, University of York, York, UK

## ABSTRACT

The mammalian jaw is often modelled as a third-class lever for the purposes of biomechanical analyses, owing to the position of the resultant muscle force between the jaw joint and the teeth. However, it has been proposed that in some rodents the jaws operate as a second-class lever during distal molar bites, owing to the rostral position of the masticatory musculature. In particular, the infraorbital portion of the zygomatico-mandibularis (IOZM) has been suggested to be of major importance in converting the masticatory system from a third-class to a second-class lever. The presence of the IOZM is diagnostic of the hystricomorph rodents, and is particularly well-developed in *Pedetes capensis*, the South African springhare. In this study, finite element analysis (FEA) was used to assess the lever mechanics of the springhare masticatory system, and to determine the function of the IOZM. An FE model of the skull of *P. capensis* was constructed and loaded with all masticatory muscles, and then solved for biting at each tooth in turn. Further load cases were created in which each masticatory muscle was removed in turn. The analyses showed that the mechanical advantage of the springhare jaws was above one at all molar bites and very close to one during the premolar bite. Removing the IOZM or masseter caused a drop in mechanical advantage at all bites, but affected strain patterns and cranial deformation very little. Removing the ZM had only a small effect on mechanical advantage, but produced a substantial reduction in strain and deformation across the skull. It was concluded that the masticatory system of *P. capensis* acts as a second class lever during bites along almost the entire cheek tooth row. The IOZM is clearly a major contributor to this effect, but the masseter also has a part to play. The benefit of the IOZM is that it adds force without substantially contributing to strain or deformation of the skull. This may help explain why the hystricomorphous morphology has evolved multiple times independently within Rodentia.

# INTRODUCTION

The mammalian jaw is frequently treated as a lever for the purposes of biomechanical analysis (e.g., *Crompton, 1963*; *Bramble, 1978*; *Greaves, 1978*; *Greaves, 1982*; *Greaves, 2000*; *Gingerich, 1979*; *Thomason, 1991*; *Satoh, 1998*; *Satoh, 1999*; *Spencer, 1998*; *Spencer, 1999*; *Satoh & Iwaku, 2006*; *Satoh & Iwaku, 2009*; *Davis et al., 2010*; *Druzinsky, 2010*; *Cornette et al., 2012*; *Becerra, Casinos & Vassallo, 2013*; *Santana, 2015*). More specifically, it is

Corresponding author
Philip G. Cox, philip.cox@hyms.ac.uk

frequently considered to be a third-class lever i.e., one in which the input force sits between the fulcrum and the output force (*Kerr, 2010*). In mammals, the resultant masticatory muscle force (the input force) is usually situated between the jaw joint (fulcrum) and the biting tooth (output force) and thus the comparison with a third-class lever is generally accurate. The advantage of positioning muscle force posterior to the teeth is that relatively wide gapes can be achieved and high tensile forces at the temporo-mandibular joint, which could lead to dislocation of the jaws, are avoided (*Greaves, 2000*; *Greaves, 2012*). However, the trade-off is that the mechanical advantage of a third-class lever is always less than one—that is, the output bite force will always be less than the effective muscle force.

It has occasionally been proposed that mammalian jaws do not always operate as third-class levers (*Davis, 1955*; *Turnbull, 1970*), and can in certain circumstances act as second-class levers with the output force between fulcrum and input force. In his classic work on the mammalian masticatory system, *Turnbull (1970)* suggested that the relative size and position of the masseter in many rodents (and a few ungulates) can shift the resultant of the masticatory musculature anterior to the distal cheek teeth, converting the masticatory system into a second-class lever during distal molar biting. Such an effect has even been claimed to occur in humans, with the jaw operating as a second-class lever during bites on the second and third molars (*Mansour & Reynik, 1975*). Alternatively, other authors have argued that although some parts of the muscle mass attach far forward on the jaws in rodents, the resultant muscle force is still located towards the posterior end of the jaw (*Greaves, 2012*).

In rodents, one muscle in particular has been identified as contributing to the jaw operating as a second-class lever. The infraorbital portion of the zygomatico-mandibularis (IOZM) is an anterior expansion of the deepest layer of the masseter, the zygomatico-mandibularis (ZM), which passes through the enlarged infraorbital foramen to take its origin on the lateral surface of the rostrum. The IOZM, also referred to as the maxillo-mandibularis (*Becht, 1953*; *Turnbull, 1970*) or medial masseter (*Wood, 1965*; *Woods, 1972*), is the one of the defining characters of hystricomorph rodents, but is also present in a somewhat smaller form in myomorphs, where it is found in combination with a rostral expansion of the deep masseter (*Wood, 1965*; *Cox & Jeffery, 2011*). Given its rostral origin on the skull and its mandibular insertion at the level of the premolar, *Becht (1953)* believed the function of the IOZM was to convert the jaw from a third-class lever to a second-class lever during molar biting.

This study seeks to understand the lever mechanics of the skull in the South African springhare, *Pedetes capensis*—a rodent species in which the IOZM is notably well-developed (*Offermans & De Vree, 1989*). *P. capensis* is a nocturnal, bipedal, saltatorial rodent that inhabits arid and semi-arid areas of southern Africa (*Peinke & Brown, 2003*). It is large for a rodent (3–4 kg) and feeds principally on grasses, especially the rhizomes of *Cynodon dactylon* and the tubers of *Cyperus esculentus* (*Peinke & Brown, 2006*). *P. capensis* and its sister-species *P. surdaster* are the only two extant members of the Pedetidae (*Wilson & Reeder, 2005*), a family which molecular analyses place as sister-group to the Anomaluridae (scaly-tailed flying squirrels) in the Anomaluromorpha, which itself is part of the mouse-related clade (*Fabre et al., 2012*). Given the presence of the IOZM muscle, the pedetids (and

anomalurids) have been identified as being hystricomorphous (*Wood, 1965*). However, the hystricomorphy seen in the Anomaluromorpha has evolved independently from that seen in three other groups of rodents: the Ctenohystrica, the Dipodidae, and some members of the Gliridae (*Hautier, Cox & Lebrun, 2015*). Thus, the function of the IOZM is of prime interest to understanding the evolution of the rodents—why has this muscle arisen independently so many times throughout rodents?

The aim of this study is to model the masticatory system of *P. capensis* to determine if it functions as a second or third-class lever, and to assess the function of the masticatory muscles, particularly the IOZM. There are two specific hypotheses that will be tested:

(1) It is hypothesised that a model of the skull of *P. capensis* will demonstrate the masticatory system operating as a second-class lever along most of the molar tooth row. This is expected based on previous dissection work by *Offermans & De Vree (1989)* who showed that a great deal of the masticatory musculature is situated alongside or anterior to the cheek teeth. The masticatory system will be determined to be a second-class lever when the bite force exceeds the effective muscle force, i.e., when the mechanical advantage is greater than one, and when the reaction force at the temporo-mandibular joint is negative.

(2) It is hypothesised that the IOZM muscle has a major role in converting the masticatory system from a third to a second-class lever in *P. capensis*. This hypothesis was previously proposed by *Becht (1953)* and is also expected owing to the large size and rostral position of the IOZM (*Offermans & De Vree, 1989*; *Offermans & De Vree, 1993*). The function of the IOZM will be determined by virtual ablation analyses i.e., removing it and other muscles from the models to elucidate the effect on the biomechanical performance of the system, as determined by mechanical advantage, principal strains and the overall deformation of the skull during biting.

Previous studies of the lever mechanics of the mammalian masticatory system have tended to focus on the mandible (*Greaves, 1978*; *Greaves, 1982*; *Greaves, 2000*; *Spencer, 1998*; *Spencer, 1999*), owing to its relatively simple shape and because its function is largely limited to feeding. However, in this study, the skull will be analysed, because of the interest in the IOZM, which is a particularly unusual fan-shaped and convergent muscle, originating on the rostrum. To address the hypotheses and to study the function of the springhare skull during biting, finite element analysis (FEA) will be employed. FEA is an engineering technique for predicting stress, strain and deformation in an object during loading (*Rayfield, 2007*), and is now frequently applied to reconstructions of skulls and other skeletal elements in order to analyse vertebrate biomechanics (e.g., *Richmond et al., 2005*; *Kupczik et al., 2007*; *Dumont et al., 2011*; *Ross et al., 2011*; *Cox et al., 2012*; *Cox, Kirkham & Herrel, 2013*; *O'Hare et al., 2013*; *Porro et al., 2013*; *Figueirido et al., 2014*; *Cuff, Bright & Rayfield, 2015*; *Sharp, 2015*; *McIntosh & Cox, 2016*; *McCabe et al., 2017*; *Tsouknidas et al., 2017*). As well as simulating stress and strain distributions, FEA is also able to predict reaction forces, and so will be used here to estimate bite force, jaw joint reaction force and mechanical advantage. Although these metrics could in theory be estimated via simple 2D lever models, it has been shown that such simplification leads to inaccuracies in muscle attachment areas, force magnitudes and directions of pull (*Davis et al., 2010*; *Greaves,*

*2012*). The advantage of FEA is that muscle forces can be distributed across the whole attachment site rather than being modelled as originating from a single centroid point, and muscle force vectors can act in three dimensions rather than two.

## MATERIALS AND METHODS

### Sample and model creation

The cranium and mandible of an adult specimen of *Pedetes capensis*, the South African springhare, were obtained from the University Museum of Zoology, Cambridge (catalogue number E.1446). The sex of the specimen was unknown, but sexual dimorphism has not been reported in this species (*Offermans & De Vree, 1989*; *López-Antoñanzas, 2016*). The specimen was microCT scanned on the X-Tek Metris system in the Medical and Biological Engineering group, University of Hull. Voxels were isometric with dimensions of 0.052 mm and 0.041 mm for the cranium and mandible respectively.

A virtual reconstruction of the cranium was created from the scan using Avizo 8 (FEI, Hillsboro, OR, USA). Bone and teeth were segmented as separate materials, but no differentiation was made between cortical and trabecular bone, nor between different materials within the teeth. These simplifications of the model geometry were felt to be justified as several previous studies have indicated that, whilst absolute strain magnitudes are impacted by the presence or absence of trabecular bone and different tooth materials, the large-scale patterns of deformation are relatively insensitive to such changes (*Fitton et al., 2015*; *Toro-Ibacache et al., 2016*). In order to reduce solution times and allow effective manipulation of the model in the FE software, the reconstruction was down-sampled to a voxel size of $0.21 \times 0.21 \times 0.21$ mm. The cranial reconstruction was then converted into a mesh of 2,310,268 eight-noded, cubic (first-order) elements via direct voxel conversion, implemented in VOX-FE, custom-built open-source FE software (*Liu et al., 2012*). The Avizo reconstruction and VOX-FE model are both available for download at https://figshare.com/articles/Springhare_FEA/5082598.

### Material properties, constraints and loads

Material properties were assigned to the model based on previous nano-indentation work on rodent skulls (*Cox et al., 2012*). Both bone and teeth were assumed to be linearly elastic and isotropic with Young's moduli of 17 and 30 GPa respectively and a Poisson's ratio of 0.3 for both. The model was constrained at both temporo-mandibular joints as well as the biting tooth. The jaw joints were constrained in all three dimensions, whilst the bite points were only constrained in the bite direction (i.e., orthogonal to the occlusal plane). This configuration of constraints is somewhat conservative (it restricts the jaw to simple hinge movements), but has been used by a number of other authors previously (*Porro et al., 2013*; *Cuff, Bright & Rayfield, 2015*; *Cox, Rinderknecht & Blanco, 2015*) and provides robust conclusions with regard to mechanical advantage. The number of nodes constrained at each location varied between 158 and 332.

Loads were applied to both sides of the model to simulate the following jaw-closing muscles (see Fig. 1) based on previous published data (*Offermans & De Vree, 1989*; *Offermans & De Vree, 1993*): masseter (combining the superficial and deep layers),

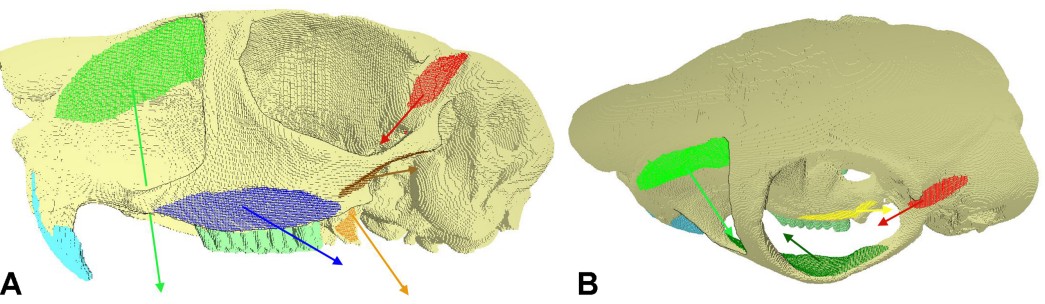

**Figure 1  FE model showing muscle attachment sites and vectors.** Skull of *Pedetes capensis* shown in (A) lateral and (B) dorso-lateral view. Key: blue, masseter; brown, posterior masseter; dark green, ZM; light green, IOZM; red, temporalis; orange, medial pterygoid; yellow, lateral pterygoid.

**Table 1  Muscle forces applied to each side of the model.** PCSA and percentage activations of each muscle from *Offermans & De Vree (1993)*.

| | PCSA (cm²) | Maximum force (N) | % activation | | Applied force (N) | |
|---|---|---|---|---|---|---|
| | | | Incision | Mastication | Incision | Mastication |
| Masseter | 2.886 | 86.58 | 20 | 70 | 17.32 | 60.61 |
| Posterior masseter | 0.654 | 19.62 | 0 | 40 | 0.00 | 7.85 |
| ZM | 3.360 | 100.80 | 60 | 100 | 60.48 | 100.80 |
| IOZM | 2.244 | 67.32 | 100 | 60 | 67.32 | 40.39 |
| Temporalis | 0.516 | 15.48 | 0 | 30 | 0.00 | 4.64 |
| Medial pterygoid | 1.130 | 33.90 | 15 | 90 | 5.09 | 30.51 |
| Lateral pterygoid | 0.519 | 15.57 | 60 | 70 | 9.34 | 10.90 |

posterior masseter, ZM, IOZM, temporalis, medial pterygoid and lateral pterygoid. Unfortunately, the superficial and deep masseters could not be modelled separately, because they were recorded as a single entity in *Offermans & De Vree (1993)*. Muscle attachment sites were determined based on the detailed descriptions and figures in *Offermans & De Vree (1989)*. Muscle directions of pull (assuming a gape angle of 0°, i.e., teeth in occlusion) were assigned using landmarks recorded from the insertion areas on a reconstruction of the springhare mandible, created from the previously gathered microCT scans. Muscle forces were calculated by multiplying the physiological cross-sectional areas (PCSA) given in *Offermans & De Vree (1993)* by an intrinsic muscle stress value of 0.3 Nmm$^{-2}$ (*Van Spronsen et al., 1989*; *Sharp, 2015*; *Tseng & Flynn, 2015*). These muscle forces were then modified based on the maximum percentage activations recorded by electromyography during incision and mastication of groundnuts (*Offermans & De Vree, 1993*). Thus, the relative proportions of total muscle force provided by each muscle were different in incisor biting to premolar/molar biting. Applied muscle forces for incision and mastication are given in Table 1. In order to ascertain the function of the masticatory muscles, versions of the model were created without each of the muscles in turn. The loaded FE model is shown in Fig. 1.

## Model solution and analysis

The model was solved for biting at each tooth along the dental arcade. Based on experimental work by *Offermans & De Vree (1990)*, all bites were modelled as bilateral i.e., the muscles on both sides of the skull were active with identical forces and the same tooth was loaded on each side of the dental row. Reaction forces at the biting tooth and at the jaw joints were calculated for each loadcase. Bite forces were divided by the effective muscle force (equal to the sum of the bite force and joint reaction forces) to calculate the mechanical advantage of the masticatory system at each tooth. As a ratio, the mechanical advantage provides a useful metric for comparing loadcases with different input muscle forces. It should be noted that it is a different measure to the mechanical efficiency of biting used in other studies (*Dumont et al., 2011*; *Cox et al., 2012*; *Cox, Kirkham & Herrel, 2013*), which divides the bite force by the total adductor muscle force, but does not take into account the orientation of muscle vectors. The distribution of maximum ($\varepsilon_1$: predominantly tensile) and minimum ($\varepsilon_3$: predominantly compressive) principal strains across the skull were examined using contour maps. Geometric morphometric methods were used to analyse deformation patterns across the skull (*Cox et al., 2011*; *Cox, Kirkham & Herrel, 2013*; *O'Higgins et al., 2011*; *McIntosh & Cox, 2016*). A set of 46 3D landmark co-ordinates (described in Fig. 2 and Table S1), based on that used in *Cox, Kirkham & Herrel (2013)*, was recorded from each solved model as well as from the original unloaded model. As changes in size are of equal significance to changes in shape during mechanical loading, the landmark sets were subjected to a Procrustes size and shape analysis (*O'Higgins & Milne, 2013*), not a Procrustes form analysis, which gives a lower weighting to size (*Fitton et al., 2015*). This was followed by a principal component analysis (PCA). All analyses were implemented in the EVAN toolbox software (www.evan-society.org).

## RESULTS

The absolute bite forces and joint reaction forces predicted by the model during biting at each tooth in *P. capensis* are given in Table 2. In addition, the mechanical advantage of the jaws at each bite has been calculated. It can be seen that joint reaction forces are negative and mechanical advantage exceeds one at all three molar teeth. In addition, the mechanical advantage is almost one (0.99) and the joint reaction force is close to zero (2.8 N) at the premolar.

The effect of removing each of the masticatory muscles on the overall mechanical advantage is given in Table 2 and shown in Fig. 3. Removal of either the IOZM or the masseter causes a decrease in mechanical advantage during both incision and mastication, with removal of the IOZM leading to the greatest decrease. Removal of the medial pterygoid muscle leads to an increase in mechanical advantage across all cheek teeth, but little effect is seen during incisor biting. Removal of the ZM causes a substantial drop in bite force at all teeth, but has little effect on the mechanical advantage of the system, except at the incisor where mechanical advantage increases in the absence of the ZM. Removal of the posterior masseter, temporalis or lateral pterygoid results in very little change in either bite force or mechanical advantage at any of the teeth, and hence the results of the models lacking these

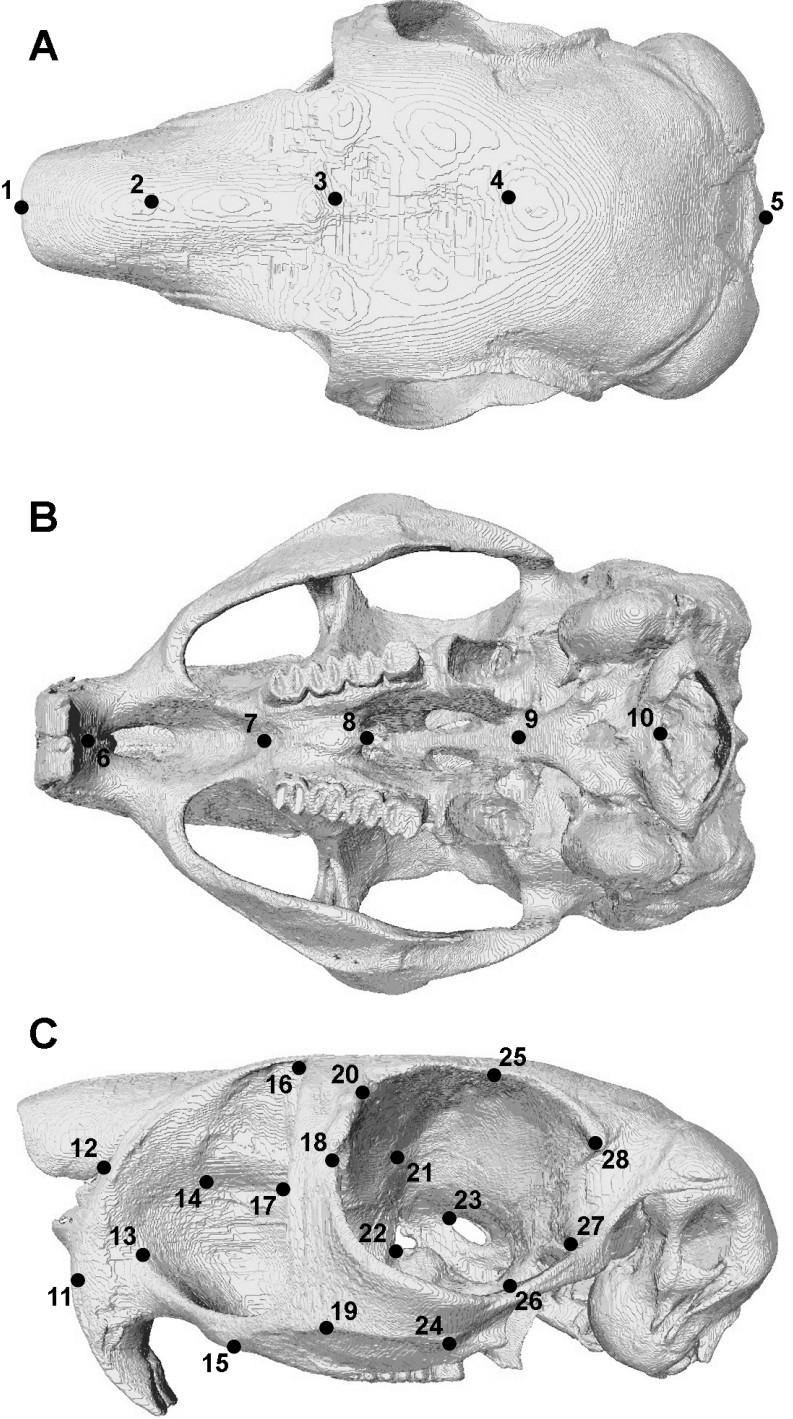

**Figure 2 Landmarks used in GMM analysis of skull deformations.** Reconstruction of skull of *Pedetes capensis* in (A) dorsal, (B) ventral and (C) left lateral view. Landmarks 11–28 recorded on both sides of the skull. Landmark descriptions are given in Table S1.

**Table 2  Bite force, joint reaction force and mechanical advantage of each model.**

| | All muscles | No masseter | No posterior masseter | No ZM | No IOZM | No temporalis | No medial pterygoid | No lateral pterygoid |
|---|---|---|---|---|---|---|---|---|
| **Bite force** | | | | | | | | |
| I | 154.6 | 138.3 | 154.3 | 122.5 | 49.9 | 154.3 | 152.0 | 154.3 |
| PM | 347.5 | 244.4 | 345.7 | 247.4 | 232.5 | 346.6 | 319.9 | 348.7 |
| M1 | 395.6 | 279.0 | 393.5 | 280.7 | 265.7 | 394.6 | 362.5 | 397.4 |
| M2 | 457.7 | 323.0 | 455.2 | 324.7 | 307.8 | 456.4 | 418.8 | 459.8 |
| M3 | 539.6 | 380.8 | 536.7 | 382.9 | 362.7 | 538.1 | 494.1 | 541.9 |
| **Joint reaction force** | | | | | | | | |
| I | 85.9 | 83.0 | 86.2 | 39.8 | 56.0 | 86.2 | 79.3 | 154.3 |
| PM | 2.8 | 29.2 | 3.4 | −2.3 | 38.8 | −0.4 | −26.9 | 9.3 |
| M1 | −45.3 | −5.4 | −44.4 | −35.7 | 5.5 | −48.3 | −69.6 | −39.5 |
| M2 | −107.3 | −49.4 | −106.1 | −79.7 | −36.4 | −110.2 | −125.9 | −102.0 |
| M3 | −189.2 | −107.1 | −187.5 | −137.9 | −91.3 | −191.8 | −201.1 | −184.0 |
| **Mechanical advantage** | | | | | | | | |
| I | 0.64 | 0.62 | 0.64 | 0.75 | 0.47 | 0.64 | 0.66 | 0.50 |
| PM | 0.99 | 0.89 | 0.99 | 1.01 | 0.86 | 1.00 | 1.09 | 0.97 |
| M1 | 1.13 | 1.02 | 1.13 | 1.15 | 0.98 | 1.14 | 1.24 | 1.11 |
| M2 | 1.31 | 1.18 | 1.30 | 1.33 | 1.13 | 1.32 | 1.43 | 1.28 |
| M3 | 1.54 | 1.39 | 1.54 | 1.56 | 1.34 | 1.55 | 1.69 | 1.51 |

muscles have not been illustrated in Fig. 3 (although the numerical data is still available in Table 2).

The contour maps of principal strain distribution across the cranium of *P. capensis* during biting on the incisor and first molar are shown in Fig. 4. It can be seen that the highest maximum and minimum principal strains are concentrated in similar areas of the skull—along the zygomatic arch and up its wide ascending ramus, and across the orbital wall, especially the anterior part. However, there are some differences between the strain distributions. The ascending ramus of the zygomatic arch is subject to greater $\varepsilon_1$ strains than $\varepsilon_3$ strains, and thus is predominantly under tension, whereas the orbital wall seems to be experiencing greater $\varepsilon_3$ strains and is likely mostly in compression. Strains are generally greater during molar biting than incision, and there is an overall caudal shift of the most highly strained regions away from the rostrum towards the orbit as the bite point moves posteriorly along the tooth row.

Figure 4 also shows the effect of removing three of the masticatory muscles (IOZM, masseter and ZM) on principal strain distributions. Despite being relatively large muscles, the impact of removing the IOZM or the masseter appears to be minimal. There are very few differences between models with all masticatory muscles applied and those without the IOZM, except for a slight reduction in strain on the rostrum and in the posterior part of the orbit during incisor and molar biting. Removal of the masseter has little effect on the strains generated by incisor biting, but reduces strains across the zygomatic arch and in the anterior part of the orbit during molar biting. Elimination of the ZM from the model,
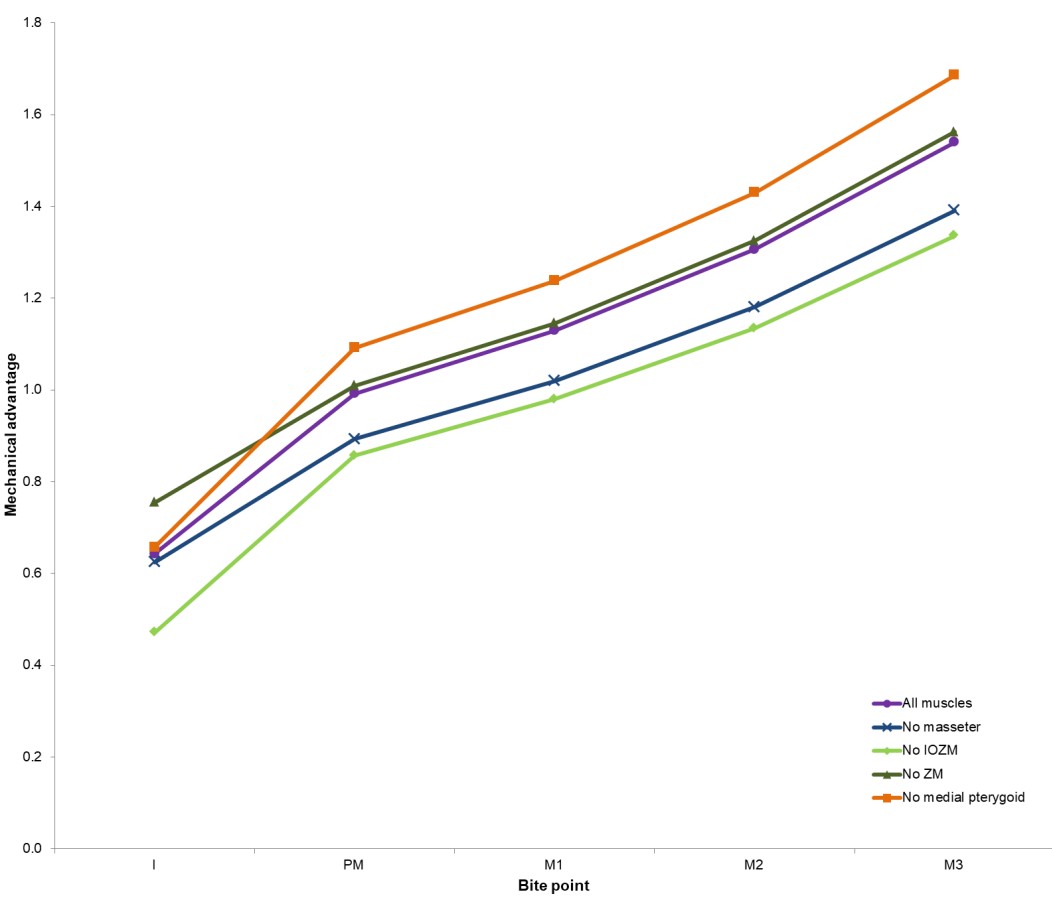

**Figure 3** **Mechanical advantage at each tooth predicted by FE model.** Abbreviations: I, incisor; PM, premolar; M1, first molar; M2, second molar; M3, third molar. Data for models with posterior masseter, temporalis and lateral pterygoid removed available in Table 2 but not illustrated here.

however, leads to a substantial reduction in $\varepsilon_1$ and $\varepsilon_3$ strains across the skull during bites at all teeth.

The geometric morphometric analysis highlights differences in the magnitude and mode of deformation between the different loadcases solved in this study. Figure 5 shows the scatter plot of the first two principal components. The first principal component encompasses 90% of the variation, and the second principal component 9% of the variation. It should be noted that to be able to visualise change across PC2, the axes have not been shown to the same scale. As demonstrated by the warped reconstructions in Fig. 5, the shape change along PC1 is mainly bending of the zygomatic arch, and this axis mostly separates loaded models from the unloaded skull, incisor bites from bites on other teeth, and models with different muscles excluded from one another. In general, incisor bites result in smaller deformations than cheek tooth bites (that is, the incisor bites are found closer to the unloaded model on PC1), whereas premolar and molar bites produce very similar deformations. Models lacking the IOZM, temporalis, medial pterygoid or lateral pterygoid deform in a very similar manner to the models with all masticatory muscles, whereas removal of the posterior masseter reduces the magnitude of deformation very

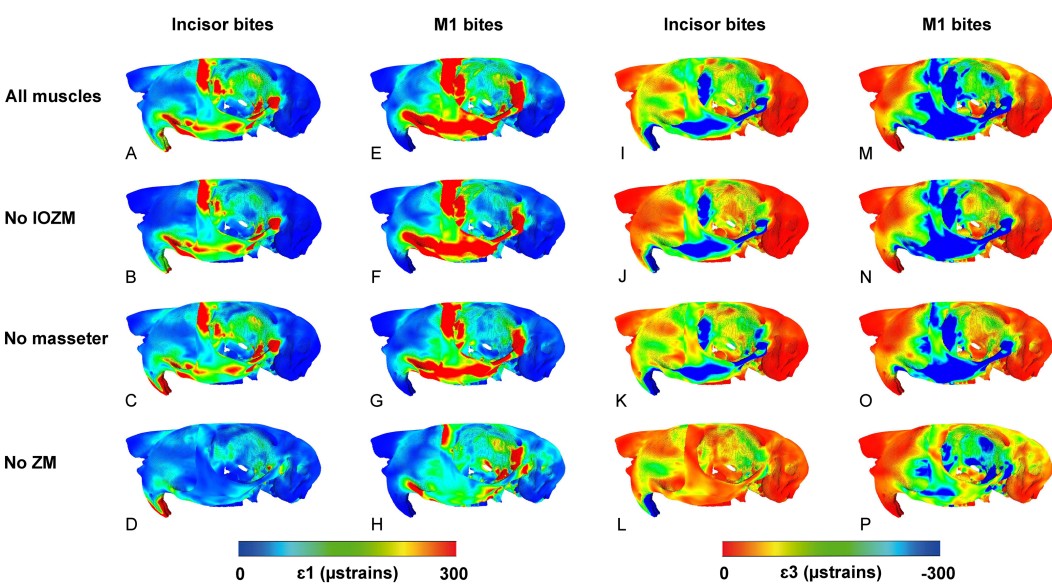

**Figure 4  Predicted principal strains across the skull of *Pedetes capensis* during incisor and first molar biting.** (A–H) maximum ($\varepsilon_1$) principal strains during incisor (A–D) and M1 (E–H) biting; (I–P) minimum ($\varepsilon_3$) principal strains during incisor (I–L) and M1 (M–P) biting. (A, E, I, M) model with all masticatory muscles included; (B, F, J, N) model with IOZM excluded; (C, G, K, O) model with masseter excluded; (D, H, L, P) model with ZM excluded.

slightly. Removal of the masseter causes a greater reduction in cranial deformation and elimination of the ZM (the largest masticatory muscle) causes the largest reduction in deformation. Shape change along PC2 represents dorso-ventral bending of the skull and separates the four different bites along the cheek tooth row.

## DISCUSSION

The results of this study support both of the hypotheses proposed here. The skull of *Pedetes capensis* operates as a second-class lever during biting along almost all of the cheek teeth (first hypothesis), and this effect can be largely ascribed to the presence of the IOZM muscle (second hypothesis), although the masseter is important in this regard as well.

### Second-class vs third-class lever

The FE model of *P. capensis* indicates that the mechanical advantage of the masticatory system is greater than one and the reaction forces at the temporo-mandibular joints are negative during bites on all three molars. Furthermore, the mechanical advantage is almost one and the joint reaction force is very close to zero during premolar biting. Thus, as the bite point moves distally along the tooth row, the system switches from a third-class to a second-class lever somewhere between the premolar and first molar. In an analysis of the mandibles of two murid species, *Apodemus speciosus* and *Cleithrionomys rufocanus*, such an effect was calculated to occur between the first and second molars (*Satoh, 1999*). The more anterior position of the effective muscle force in the springhare may be driven in large part by its unusual cranial morphology. In most hystricomorph rodents, the anterior root of the zygomatic arch arises from the skull approximately at the level of the first cheek tooth, but

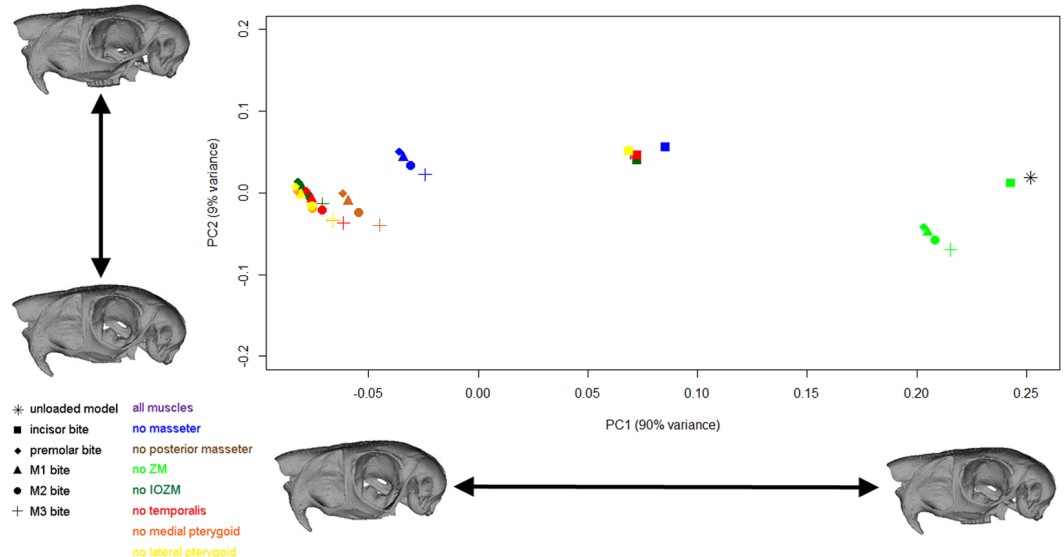

**Figure 5  GM analysis of cranial deformations in *Pedetes capensis*.** Plot of the first two principal components from a GM analysis of 46 landmarks and 41 models. Axes not to scale. Cranial reconstructions indicate shape changes (x400) along axes.

in *P. capensis* it is much further forward, attaching to the shortened rostrum just posterior to the upper incisor (*Offermans & De Vree, 1989*). Thus, the masticatory musculature, as a whole, is more rostrally positioned than in most other rodents, and so the jaw becomes a second-class lever at more anterior position along the tooth row.

The prediction of large tensile forces at the jaw joints of the springhare is a result that is at odds with some published work on mammalian masticatory biomechanics. It has been argued that the capsule and ligaments of the mammalian jaw joint are poorly suited to resisting high tensile forces (*Greaves, 2000*; *Greaves, 2012*), and that the mammalian masticatory system has evolved to maintain the resultant muscle force within the posterior third of the jaw ramus in order to prevent instability and dislocation of the jaws (*Greaves, 1978*; *Greaves, 1982*; *Greaves, 2000*; *Spencer, 1998*; *Spencer, 1999*). The mismatch between the results presented here and this earlier work may be the product of insufficient knowledge of muscle recruitment in springhare mastication and also the limitations of static FE models. Although the muscle forces were based on *in vivo* work that measured the degree to which each muscle was activated during biting (*Offermans & De Vree, 1993*), a single set of values was used for all molar bites; the only variation in muscle recruitment was between incisor and cheek teeth bites. It has been shown that muscle recruitment can vary a great deal from tooth to tooth, and even between bites on the same tooth (*Cleuren, Aerts & De Vree, 1995*). Furthermore, the percentage activations used to calculate muscle force are the maximum applied at any point during the masticatory cycle. Obviously, the recruitment of each muscle changes as the teeth are brought into and out of occlusion, but a static model cannot reflect this. Thus, the results here indicate that jaw is a second-class lever in molar biting, but this only holds true for the muscle recruitment pattern applied to the model. In

reality, the springhare may modulate fibre recruitment within its complex set of muscles to maintain the jaws as a third-class lever even at the distal molars. For instance, the external pterygoid may increase in activation during distal molar biting to resist dislocation of the jaws as has been suggested to occur in murids (*Satoh, 1999*).

Even after taking the limitations of the model into consideration, it is clear that the springhare has the potential to generate very high bite forces at its molar teeth. Moreover, even if not quite a second-class lever these bites are very efficient, so high bite forces can be produced without having to massively increase the overall adductor muscle mass. It is likely that *P. capensis* has evolved this highly efficient feeding system in order to cope with the demands of the arid environment in which it lives (*Peinke & Brown, 2003*). Springhares are herbivorous, feeding almost exclusively on grasses (*Peinke & Brown, 2006*). Although they are known to eat the leaves, springhares tend to favour underground storage organs, such as rhizomes and tubers, particularly during autumn and winter when nutritional reserves are transferred away from leaves and into the leaf bases and roots (*Peinke & Brown, 2006*). These storage organs tend to be mechanically demanding to eat, requiring a great deal of mastication to break down, which may have driven the evolution of the highly efficient masticatory system of springhares. The disadvantage of the masticatory arrangement seen in *P. capensis* is that the rostral position of many of the jaw-closing muscles is likely to severely limit maximum gape. However, given their preferred diet of grasses, these limitations are not likely to impact the ability of springhares to feed effectively.

## Function of the masticatory muscles

The virtual ablation experiments, in which masticatory muscles were sequentially removed from the FE model, show that the IOZM is the most important muscle in converting the masticatory system from a third-class to second-class lever in molar biting, as predicted by the second hypothesis. When the IOZM is removed, the mechanical advantage decreases, indicating that more force is being directed towards the jaw joints. This has the effect that the point at which the system switched from operating as a third-class to a second-class lever moves back to somewhere between the first and second molars. Thus, this result supports the idea proposed by *Becht (1953)* that the function of the IOZM is to convert the masticatory system to a second-class lever during molar biting, at least in *P. capensis*. Removal of the IOZM had very little impact on the distribution and magnitudes of strain across the skull (Fig. 4), nor did it greatly change the overall deformation of the skull during biting (Fig. 5), as has also been noted in another species of rodent, *Laonastes aenigmamus* (*Cox, Kirkham & Herrel, 2013*). Thus, it appears that the increase in mechanical advantage gained by the presence of an IOZM muscle does not come at the cost of greatly increased strain or deformation, either in the region of the IOZM origin or elsewhere on the skull. In addition, the development of the IOZM in *P. capensis* may be a response to the need to generate large forces at the incisors, such as during the cracking of nutshells (*Offermans & De Vree, 1990*) or gnawing of roots and tubers (*Peinke & Brown, 2006*). The anterior position of the IOZM on the skull means that improvements in incisor bite force can be achieved without an excessive increase in muscle size.

The ZM is one of the largest masticatory muscles in the springhare (*Offermans & De Vree, 1993*), which is unusual among rodents; usually the ZM is smaller than the superficial and deep masseters and the IOZM (*Turnbull, 1970*; *Woods, 1972*; *Cox & Jeffery, 2011*; *Baverstock, Jeffery & Cobb, 2013*; *Becerra et al., 2014*). Despite its large size, the removal of the ZM from the FE model had very little effect on the efficiency of the masticatory system i.e., the mechanical advantage and joint reaction force remained largely the same. Thus, by virtue of being large, the ZM is an important muscle for increasing overall bite force, but its presence does not alter the efficiency of the system a great deal. However, the ZM does have a large effect on the deformation of the springhare skull during biting. The GMM analysis showed that elimination of the ZM greatly reduces the magnitude of deformation experienced by the skull (Fig. 5), much more so than any other masticatory muscle. This appears to be a consequence of the attachment site of the ZM on the zygomatic arch. As has been found in other FEA studies of mammal skulls (*Bright, 2012*; *Cox et al., 2012*; *Fitton et al., 2012*), the morphology of the zygomatic arch makes it susceptible to larger deformations than other parts of the skull. Indeed, in this study, deformations of the zygoma overwhelm deformations in all other parts of the skull, as can be seen from the warped reconstructions in Fig. 5. The large size and location of the ZM in *P. capensis* leads to it being the principal generator of zygomatic strain and deformation. This can be seen in Fig. 4, where removal of the ZM vastly reduces strain in the zygomatic arch.

It has been suggested that the large zygomatic strains seen in many FEA studies of mammalian skulls may be artificial and the result of a failure to incorporate important soft tissue structures into the models. In particular, the temporal fascia has been shown to resist inferior bending of the zygomatic arch in an FE model of a macaque (*Curtis et al., 2011*). This is unlikely to be the case here as no temporal fascia was reported by *Offermans & De Vree (1989)* in their dissection of a springhare. Furthermore, the temporalis is extremely small in *P. capensis*, and the temporal region is positioned distinctly caudal the zygomatic arch, which would reduce the ability of a temporal fascia to counteract ventral deflection of the zygomatic arch. However, there is an aponeurosis attached extensively around the margin of the infraorbital fossa (*Offermans & De Vree, 1989*), which may have the potential to resist some bending in the anterior part of the zygomatic arc and its ascending ramus. Further work, both *ex vivo* dissection and *in silico* modelling, is necessary to understand the biomechanical consequence of this aponeurosis.

The masseter has been shown to have a similar effect to the IOZM with regard to bite force, although not quite to the same extent. It, too, appears to shift the resultant masticatory muscle force anteriorly along the rostrum, thus directing force towards the biting tooth and away from the jaw joints. Removal of the masseter has much the same effect as removing the IOZM—the mechanical advantage is decreased and the point at which the system becomes a second-class lever is shifted posteriorly along the tooth row. Unfortunately for this study, *Offermans & De Vree (1993)* did not separate the superficial and deep masseter when measuring PCSA, so the two muscles could not be modelled separately in the FEA. However, the illustrations in *Offermans & De Vree (1989)* indicate that the fibres of the superficial masseter have a more horizontal alignment than those of the deep masseter (as in most rodents, e.g., *Turnbull, 1970*), so it is likely that the superficial

masseter is the more important division of the masseter with regard to the operation of the jaw as a second-class lever. In terms of cranial deformations, the masseter has a similar, but lesser, effect to the ZM. Owing to its attachment to the zygomatic arch, the action of the masseter generates inferior bending of the arch, and thus its removal tends to reduce global deformation of the springhare cranium (Fig. 5). It can also be seen that that removal of the masseter causes a slight reduction in zygomatic and orbital strains during molar biting (Fig. 4).

The medial pterygoid, because of its posterior line of action, tends to direct muscle force away from the teeth and towards the jaw joints, unlike the IOZM and masseter. Thus removal of the medial pterygoid increased the mechanical advantage of the masticatory system. Overall, the presence of the medial pterygoid increases bite force because it increases the total input adductor muscle force, but it does so in a somewhat inefficient manner. Thus, although the medial pterygoid has a relatively large PCSA, it has a relatively small effective muscle force owing to its orientation. However, it has been shown that the medial pterygoid is important in other aspects of masticatory biomechanics, notably as a counterbalance to the lateral pull of the masseter, thereby preventing wishboning of the mandible (eversion of the lower border and angular process) and reducing tensile strains at the symphysis (*Hiiemae, 1971*; *Satoh, 1998*; *Cox & Jeffery, 2015*).

The posterior masseter, temporalis and lateral pterygoid are very small compared to the other masticatory muscles in *P. capensis*, each providing less than 11% of the total adductor muscle force. Hence, the impact of their removal on bite force and mechanical advantage was minimal. Similarly, removal of these muscles had a very limited impact on the overall deformation of the skull (Fig. 5). The models without the temporalis and lateral pterygoid can barely be distinguished from the models with all masticatory muscles. The models without a posterior masseter show a very slight reduction in the magnitude of cranial deformation. This is because the posterior masseter attaches to the caudal part of the zygomatic arch and thus is able to cause a small amount of posterior deflection. It is likely that these muscles contribute to aspects of the masticatory process other than bite force generation, especially the antero-posterior movements of the mandible relative to the skull that are common to rodents. The temporalis, whilst clearly too small to be a powerful elevator of the jaw as in myomorphs (*Hiiemae, 1971*; *Gorniak, 1977*), may have a braking role during the power stroke of chewing (*Byrd, 1981*), and the lateral pterygoid may be important in protraction of the mandible (*Weijs & Dantuma, 1975*; *Gorniak, 1977*) or in resisting tensile forces at the temporo-mandibular joint as mentioned above (*Satoh, 1999*).

## CONCLUSIONS

The masticatory system of *P. capensis* has been shown to have the potential to act as a second-class lever along the majority of the cheek tooth row and, as predicted by *Becht (1953)*, the IOZM is a particularly important muscle in the switch from third-class to second-class lever mechanics. It should be noted that masseter also plays an important role in this regard. This analysis of muscle function is, of course, specific to *P. capensis* and

further analyses of other species are necessary to determine whether the conclusions hold true for other rodents. However, the position of the IOZM, far forward on the rostrum, makes it likely that it will have some role to play in increasing the mechanical advantage of the masticatory system in most hystricomorph rodents (the exact scale of the effect being dependent on the size of the IOZM relative to the other masticatory muscles). Previous research has suggested that, amongst rodents, sciuromorphs are adapted for efficient gnawing at the incisors, whereas hystricomorphs are adapted to efficient grinding at the molars (*Cox et al., 2012*). *Druzinsky (2010)* determined that of all the masticatory muscles, it is the anterior deep masseter that confers efficacious incisor bites in sciuromorphs. Here, it is indicated that the IOZM provides efficiency in molar bites in hystricomorphs, without substantially increasing strains across the skull or the magnitude of cranial deformation. This may go some way to explaining why hystricomorphy has evolved convergently at least four times within the rodents.

## ACKNOWLEDGEMENTS

The author thanks Matt Lowe and Rob Asher (University Museum of Zoology, Cambridge) for the loan of the specimen, and Sue Taft and Michael Fagan (Medical and Biological Engineering Group, University of Hull) for microCT scanning. Thanks are due to Paul O'Higgins for help with geometric morphometrics, and to three anonymous reviewers for their useful comments that helped improve this manuscript.

### Funding

The author received no funding for this work.

### Competing Interests

Philip G. Cox is an Academic Editor for PeerJ.

### Author Contributions

- Philip G. Cox conceived and designed the experiments, performed the experiments, analyzed the data, wrote the paper, prepared figures and tables, reviewed drafts of the paper.

### Data Availability

Cox, Philip (2017): Springhare FEA. figshare.

https://doi.org/10.6084/m9.figshare.5082598.v2.

### Supplemental Information

Supplemental information for this article can be found online at http://dx.doi.org/10.7717/peerj.3741#supplemental-information.

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
