# Peer review of "The jaw is a second-class lever in Pedetes capensis (Rodentia: Pedetidae)"

_PeerJ, doi:10.7717/peerj.3741_

## Round 0.1 · original submission · Minor Revisions

Dear Phil,

Although the comments of the reviewers were very positive, you must address all of the reservations and criticisms of the reviewers before the manuscript is accepted. Please note that Reviewer 3 provided an annotated version of the manuscript with the review.

Thanks for submitting to PeerJ.

Best,
Robert

Reviewer 1 ·

Basic reporting

Excellent in all regards.

Experimental design

I see no problems with the experimental design. Overall, an thorough and thoughtful manuscript. A minor concern is that the jaw joints may be over constrained, but this is the conservative approach that strengthens the conclusions regarding mechanical advantage.

Validity of the findings

no comment

Additional comments

An interesting, well written, and robust manuscript. A pleasure to read.

Reviewer 2 ·

Basic reporting

The reviewed manuscript is well-written and presents meaningful results on an interesting topic. I believe it to be worthy of publication, once the author addresses some points, which in my opinion require rectification.
The manuscript addresses a topic, in a scientific area that is of interest to several research groups worldwide. Jet a large proportion of the recent referenced literature (only 36 of which was elaborated over the last 5 years) stems from the author’s own group! I believe that updating some of the references to more recent ones and including some further studies done recently within other teams, would be of benefit to the manuscript’s novelty.

Experimental design

The Finite Element model description is detailed as to the materials section and boundary conditions, but deficient as to the verification/validation of the model. Below are some aspects that should be, in my opinion, addressed:
• Voxel based meshes are generally in-efficient as to their computational effort, unless the resulting grid is further optimized. This is in my experience tricky, as mesh size is driven by scanner accuracy (voxel size) and not indicative of the grid’s quality. Please provide some further details on the element’s quality criteria (size, aspect ratio etc.)
• The criteria of identifying the optimum mesh density are associated to results accuracy in terms of verification, thus addressing the soundness of the theoretical model. This is usually achieved with a mesh independent grid, ensuring that coarsening of the mesh does not disturb the stress field by more than 2% (Zienkiewicz OC and Taylor RL. 1989. The Finite Element Method, McGraw-Hill, New York.). Were such considerations taken into account?
• It would be also helpful to provide some information about element type (first or second order elements, integration etc.) that were used during the analyses as these can also affect results accuracy.

Validity of the findings

The author argues a valid point, that the jaw in rodents is modeled more accurately by second- rather than third-class lever (during distal molar bites). Although I’m inclined to agree, I cannot but wonder, why the author focuses solely on the cranium to prove his hypothesis. A recent study [Front. Physiol. 8:273. doi: 10.3389/fphys.2017.00273] demonstrates stress distributions in the jaw, that sustain the author’s argument (e.g. incisal biting induces stress concentrations at the TMJ as would be expected from a third-class lever, whereas chewing results in biomechanical response indicative of a second-class lever). I believe that a model also considering jaw morphology and musculature, would be clearly better suited for the purpose of this study. I would, in these terms, urge the author to at least discuss his findings with respect to literature considering mandible models, as to strengthen his hypothesis.

Reviewer 3 ·

Basic reporting

I find the text mostly clear and easy to read, in a proper language. Figures and tables summarize and reinforce the overall understanding (see further comments and suggestions below).

Experimental design

The research seems to be really interesting, well focused and meaningful for the understanding of the masticatory biomechanics. It also implements different methodologies to understand the case in a holistic way, combining finite elements analysis and geometric morphometrics on the skulls. Nevertheless, I would be a bit concerned about using the modelling to assess both the skull deformation and the final output (i.e. bite force) at the same time, instead of gathering the in vivo bite force and using it to set a proper model (see further comments below). Moreover, not all the force produced by a muscle is contributing to the biting event due to the tridimensional orientation of the muscle and its line of action (in some cases, as for the superficial masseter, the muscle’s orientation and line of action are different). Therefore, I would suggest to the author using the effective muscle force instead of the input muscle force to improve the quality of the analysis (see further comments below).

Validity of the findings

I find the analysis robust and consistent; despite a few conclusions on the mechanical functioning, as simplifying a conversion from a third-class lever to a second-class lever without considering the chance that the muscles input to the biting could be modulated depending on each case (see further comments below).

Additional comments

Please find all my comments and suggestions, by lines, in the attached file.

Annotated reviews are not available for download in order to protect the identity of reviewers who chose to remain anonymous.

---

## Round 0.2 · Minor Revisions

I believe that the manuscript is almost ready to accept. Please address the minor editorial changes suggested by Reviewer 3. Also, please remember that there will be no proof reading of your manuscript by the PeerJ staff, so check that your manuscript is ready for publication when you re-submit.

Thanks for your hard work.

Regards,
Robert

Reviewer 1 ·

Basic reporting

no comment

Experimental design

no comment

Validity of the findings

no comment

Reviewer 2 ·

Basic reporting

The manuscript is well argued and written.

Experimental design

The Experimental design is sound, while some minor concerns I had with the initial submission were addressed.

Validity of the findings

I believe the current version of the manuscript contributes to the stat of the art in the field.

Additional comments

The author has addressed all my comments and I believe this version of the manuscript to be worthy of publication.

Reviewer 3 ·

Basic reporting

The manuscript has been highly improved in both the text and the figures, and it seems now clear and straight forward.

Experimental design

By acknowledging the diversity of potential explanations beyond the ones found in this research, I find the author’s ideas and conclusions much more realistic than how they were presented in the previous version, and better for giving the chance to launch new lines of investigation that will let us ultimately understand the whole masticatory apparatus.

Validity of the findings

No more comments.

Additional comments

I would like to suggest only a few more and very minor changes (see comments in the attached file; line numbers refer to this new version in pdf).

Annotated reviews are not available for download in order to protect the identity of reviewers who chose to remain anonymous.

---

## Round 0.3 · accepted · Accept

Thank you again for your submission to PeerJ.